# ADACAT: Adaptive Categorical Discretization for Autoregressive Models

Qiyang Li[1]            Ajay Jain[1]            Pieter Abbeel[1]

[1] University of California Berkeley, Berkeley, CA, USA

## Abstract

Autoregressive generative models can estimate complex continuous data distributions, like trajectory rollouts in an RL environment, image intensities, and audio. Most state-of-the-art models discretize continuous data into several bins and use categorical distributions over the bins to approximate the continuous data distribution. The advantage is that the categorical distribution can easily express multiple modes and are straightforward to optimize. However, such approximation cannot express sharp changes in density without using significantly more bins, making it parameter inefficient. We propose an efficient, expressive, multimodal parameterization called Adaptive Categorical Discretization (ADACAT). ADACAT discretizes each dimension of an autoregressive model adaptively, which allows the model to allocate density to fine intervals of interest, improving parameter efficiency. ADACAT generalizes both categoricals and quantile-based regression. ADACAT is a simple add-on to any discretization-based distribution estimator. In experiments, ADACAT improves density estimation for real-world tabular data, images, audio, and trajectories, and improves planning in model-based offline RL.

## 1 INTRODUCTION

Deep generative models estimate complex, high-dimensional distributions from samples. Autoregressive models like NADE [Larochelle and Murray, 2011, Uria et al., 2016], PixelRNN [Van Oord et al., 2016] and GPT [Radford et al., 2018] express a joint distribution by decomposing it into a product of simpler one-dimensional conditionals. Each of these conditionals $p(x^t|x^1, x^2 \ldots x^{t-1})$ is parameterized by a neural network

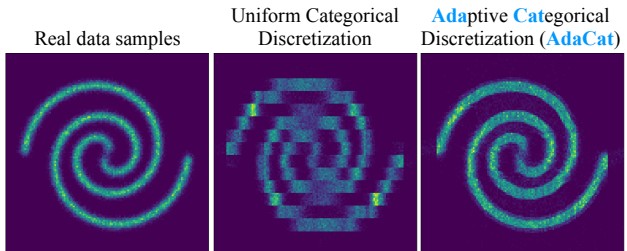

Real data samples | Uniform Categorical Discretization | Adaptive Categorical Discretization (AdaCat)

Figure 1: In the 2-D two-spirals dataset, an autoregressive model parameterizes $p(x^1)$, the marginal distribution over the first dimension, and $p(x^2|x^1)$ a conditional distribution over the second. Uniform discretization (middle) divides their 1-D support into 16 equal-sized intervals and parameterizes each conditional with a categorical. However, it poorly fits the continuous samples. In contrast, parameterizing $p(x^t|x^{<t})$ with ADACAT closely approximates the target distribution with the same number of bins.

mapping from a subset of observed variables to logits over the next dimension. For discrete data like language tokens, the conditional takes the form of a categorical distribution. Categorical distributions are relatively easy to optimize, flexible and can easily express multimodal distributions as each bin's logit is independently predicted.

Ordinal and continuous data such as image intensities ranging from 0 to 1 have a natural ordering between possible values of each dimension $x^t$. The categorical does not exploit this ordering, instead separately predicting each bin. Categorical distributions also scale poorly when encoding highly precise data like agent trajectories, tabular datasets and audio [Oord et al., 2016]. Auditory quality degrades if the waveform is quantized to less than 8-16 bits (256-65k intensity levels). Control applications often need high precision as well. Unfortunately, categorical likelihood degrades rapidly at high quantization levels. The uniformly discretized model in Figure 1 has a negative log-likelihood $-0.85$ with 16 bins, while our adaptively discretized approach achieves NLL $-1.02$ with the same architecture and

*Accepted for the 38th Conference on Uncertainty in Artificial Intelligence* (UAI 2022).

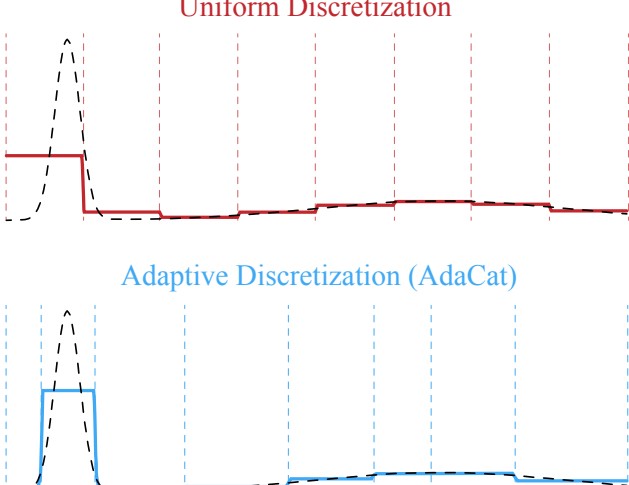

Uniform Discretization

Adaptive Discretization (AdaCat)

Figure 2: ADACAT learns how to discretize the support of continuous distributions for expressive, parameter efficient density estimation, and generalizes other discretization strategies like categoricals with equal bin widths (top). The flexibility afforded by adaptive discretization allows closer approximations of target densities, such a mixture of two Gaussians with different scales (bottom).

number of bins (lower is better). We note that the negative log-likelihood can be negative as it is computed on the continuous density by treating the discretized distribution as a mixture of uniform distributions. Halving the width of a particular bin in the categorical distribution would require double the parameters in the final layer of the network.

Past work tries to improve the efficiency of categoricals with hierarchical softmax [Morin and Bengio, 2005] or quantile-based discretization [Janner et al., 2021]. Heuristic, hand-engineered discretizations like the $\mu$-law [Oord et al., 2016] reduce quantization error and improve perceptual quality. As an alternative, a single Gaussian, Gaussian mixtures [Bishop, 1994] or logistic mixtures [Salimans et al., 2017] are frequently used for parameter efficient conditionals, but can be hard to optimize, especially as the number of mixture components increases.

In this work, we propose a parameterization of 1-D conditionals that is parameter efficient, expressive and multimodal. We propose Adaptive Categorical Discretization (ADACAT). Based on the observation that high precision is often only required in a small subset of a distribution's support, ADACAT is a distribution parameterized by a vector of interval masses *and* interval widths. ADACAT is depicted in Figure 2. In contrast to categoricals with equal bin widths, variable bin widths allow the network to localize mass precisely without increasing precision elsewhere. Compared to non-uniform but fixed discretizations like quantiles, ADACAT parameters are adaptive: they are predicted by a neural network conditioned on prior dimensions, which is impor-

tant as the best discretization for a particular conditional differs from the best for the marginal.

We also propose an analytic target smoothing strategy to ease optimization, and draw connections between target smoothing and dequantization [Uria et al., 2016] and score matching [Vincent, 2011]. In experiments, ADACAT with target smoothing scales better to few parameters or is competitive with strong baselines on image density estimation, offline reinforcement learning, tabular data and audio.[1]

## 2 ADAPTIVE CATEGORICAL DISCRETIZATION

### 2.1 ADACAT DISTRIBUTION

The ADACAT distribution is a particular subfamily of mixtures of uniform distributions where each mixture component has non-overlapping support. A standard ADACAT distribution $\text{ADACAT}_k(w, h)$ has $k$ components with a support over $[0, 1)$. It is parameterized by two vectors in the $k$-dimensional simplex: $w, h \in \Delta^{k-1}$. Thus, $w$ and $h$ are normalized. $w$ is additionally constrained to be non-zero in all of its elements. The probability density function (PDF) of an ADACAT distribution is defined as,

$$f_{w,h,k}(x) = \sum_{i=1}^{k} \left\{ \mathbb{I}_{[c_i \leq x < c_i + w_i]} \frac{h_i}{w_i} \right\} \tag{1}$$

where $c_i = \sum_{j=1}^{i-1} w_j$ is the prefix sum of the dimensions of parameter $w$ and $\mathbb{I}_{[\cdot]}$ is the indicator function.

Intuitively, $w_i$ captures the size of each discretized bin (support of each mixture component), $h_i$ captures the probability mass assigned to each bin, and $\frac{h_i}{w_i}$ is the density contributed by each bin.

### 2.2 RELATIONSHIP WITH UNIFORM AND QUANTILE DISCRETIZATION

**Connection to Uniform Discretization** Generative models over ordinal data like PixelRNNs [Van Oord et al., 2016] commonly divide the support of 1-D distributions into equal-width bins, and share the same bins across all dimensions of the data. This allows neural networks to parameterize the distribution with a simple classification head that predicts a categorical distribution over bins. ADACAT generalizes 1-D categorical distributions with uniformly discretized support. If $w$ is set to be $w_i = \frac{1}{k}, \forall i$, the distribution is reduced to a categorical distribution over $\{0, \frac{1}{k}, \frac{2}{k}, \cdots, \frac{k-1}{k}\}$ augmented with a uniform noise of magnitude $\frac{1}{k}$. Figure 2 shows how

---

[1]The code for reproducing the experiments in this paper is available at github.com/ColinQiyangLi/AdaCat. Website: colinqiyangli.github.io/adacat.

ADACAT is more expressive than a uniformly discretized categorical, allowing bin widths to vary and more closely approximating the modes of a mixture of two Gaussians.

**Connection to Quantile-based Discretization**  ADACAT also generalizes quantile-based discretization, which discretizes a distribution's support by binning data into groups with equal numbers of observed data points. If $h$ is set to be $h_i = \frac{1}{k}, \forall i$, *i.e.* the same mass in every bin, the vector $w$ can be interpreted as the $k$-quantile of the distribution. This strategy is employed by generative models like the Trajectory Transformer [Janner et al., 2021], which pre-computes and fixes the bin widths $w$ separately for each dimension to achieve equal mass $\frac{1}{k}$ per bin of the marginal distributions of the training set, then predicts mass $h$ with a neural network based on observed dimensions.

## 2.3 AUTOREGRESSIVE PARAMETERIZATION

In problems with dimension greater than 1, we use deep autoregressive models to factorize the joint density $f(x)$ into multiple 1-D conditional ADACAT distributions:

$$p_\theta(x) = \prod_{t=1}^{m} p_\theta(x^t | x^{<t}) = \prod_{t=1}^{m} f_{w^t, h^t, k}(x^t)$$

For each dimension, conditioned on observed or generated values of prior dimensions, the neural net $g_\theta$ outputs two unconstrained parameters $\{\phi^t, \psi^t\} = g_\theta(x^{<t})$, where $\phi^t, \psi^t \in \mathbb{R}^k$. The predicted $\phi$ and $\psi$ represent the unnormalized log values for $h$ and $w$ for each dimension. These parameters are normalized independently using a softmax to satisfy the normalization and positivity constraints:

$$w_i^t = \frac{\exp(\psi_i^t)}{\sum_{j=1}^{k} \left[\exp(\psi_j^t)\right]}, \quad h_i^t = \frac{\exp(\phi_i^t)}{\sum_{j=1}^{k} \left[\exp(\phi_j^t)\right]} \quad (2)$$

where $\phi^t = \phi_\theta(x^{<t}), \psi^t = \psi_\theta(x^{<t})$.

Unlike uniform, quantile-based, or heuristic discretization strategies, our autoregressive model can adaptively choose how to discretize each dimension's conditional distribution based on observations. Adaptivity improves expressiveness, since density can be precisely localized in regions of interest, and the discretization can vary across data dimensions. This is especially important for problems where the optimal discretization is not known *a priori*. In the 2-D dataset shown in Figure 1, fixed discretizations poorly express the inherent multimodality in the data, while ADACAT's adaptivity allows the network to shift modes of $p_\theta(x^2|x^1)$ for different values of the first dimension, $x^1$.

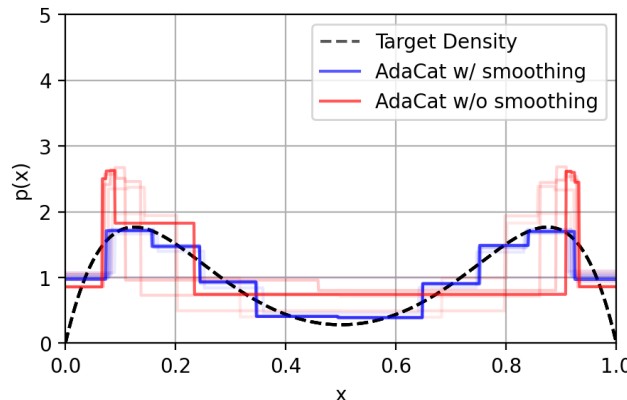

Figure 3: **1-D Toy Density Modeling**: ADACAT optimized with the non-smoothed objective (red) suffers from **bin collapse**. The non-smoothed objective shrinks the size of most bins until they are degenerate with small support in order to increase density at the modes. In contrast, with the smoothed objective (blue), ADACAT closely approximates the target. The transparent curves show an evolution of the learned density at different optimization iterations, with more transparent ones being earlier in the optimization. The code for reproducing this figure and an animated version are available in the supplement.

## 3 OPTIMIZING WITH ANALYTIC TARGET SMOOTHING

Autoregressive models with ADACAT conditionals can be estimated by minimizing the Kullback–Leibler (KL)-divergence between the target distribution and the learned density $D_{\mathrm{KL}}(p_{\mathrm{data}}(x) \| p_\theta(x))$. The KL reduces to the empirical log-likelihood objective below, where $x_1, \cdots, x_n$ are sampled from the data distribution $p_{\mathrm{data}}$:

$$\hat{\mathcal{L}}_{\mathrm{ll}} = \frac{1}{n} \sum_{d=1}^{n} \sum_{t=1}^{m} \log p_\theta(x_d^t | x_d^{<t}) \quad (3)$$

However, due to the discontinuous nature of the ADACAT density function, we observed that naïvely maximizing the empirical log-likelihood encourages the model to get trapped in poor local optima. This phenomenon is best illustrated in 1-D, as in Figure 3. The density in red is estimated with maximum likelihood $\hat{\mathcal{L}}_{\mathrm{ll}}$ (3), but the bin widths shrink over the course of optimization and reach small values. Density is overestimated in between modes and underestimated in regions where a single uniform mixture component needs to cover a large interval.

We provide one possible explanation for bin collapse. Rewriting the log-likelihood with ADACAT's PDF (1) based

on a summation over bins $i = 1$ to $k$,

$$\hat{\mathcal{L}}_{\text{ll}} = \frac{1}{n} \sum_{d=1}^{n} \sum_{t=1}^{m} \log \underbrace{\sum_{i=1}^{k} \left\{ \mathbb{I}_{\left[c_i^t \leq x < c_i^t + w_i^t\right]} \frac{h_i^t}{w_i^t} \right\}}_{f_{w^t, h^t, k}(x^t)} \quad (4)$$

Due to the constraint that mixture components are non-overlapping, only a single term of the inner summation is non-zero in (4). The loss separates into terms for $h$ and $w$,

$$\hat{\mathcal{L}}_{\text{ll}} = \frac{1}{n} \sum_{d=1}^{n} \sum_{t=1}^{m} \sum_{i=1}^{k} \mathbb{I}_{\left[c_i^t \leq x < c_i^t + w_i^t\right]} \left\{ \log h_i^t - \log w_i^t \right\} \quad (5)$$

Maximizing the loss for data point $x_d$ pushes for higher density $\frac{h_i^t}{w_i^t}$ when $x_d^t$ lies in for bin $i$ of the conditional $p_\theta(\cdot | x_d^{<t})$. This density is increased by either increasing log mass $\log h_i^t$ or decreasing log bin width $\log w_i$. For uniform and heuristic discretizers, $\log w_i^t$ is fixed. However, updating the bin width $w_i^t$ with finite step sizes can make the data point $x_d^t$ move out of the current bin discontinuously, which can result in biased gradient estimates (see the supplement C.1).

The gradient $\frac{d}{dw_i^t} \hat{\mathcal{L}}_{\text{ll}}$ is also constant for any value of a sample within bin $i$ as the density is piece-wise constant, so the gradient encouraging bin collapse does not attenuate as the sample approaches a bin boundary. Once a bin is updated to exclude a particular data point, only the normalization of $w^t$ encourages the bin to grow to include the data point again, but we empirically find that this is not enough to prevent collapse. Instead, optimization could shrink the new bin $w_{i+1}^t$ or $w_{i-1}^t$, repeating until a majority of the mixture components collapse to support a small fraction of the overall interval.

Luckily, this issue can be largely alleviated by using a smoothed objective:

$$\hat{\mathcal{L}}_s = \frac{1}{n} \sum_{d=1}^{n} \sum_{t=1}^{m} \mathbb{E}_{\zeta(\tilde{x} | x_d^t)} \left[ \log p_\theta(\tilde{x} | x_d^{<t}) \right] \quad (6)$$

$$= \frac{1}{n} \sum_{d=1}^{n} \sum_{t=1}^{m} \left[ \int_{\tilde{x}} \zeta(\tilde{x} | x_d^t) \log p_\theta(\tilde{x} | x_d^{<t}) d\tilde{x} \right] \quad (7)$$

where $\zeta(\tilde{x} | x)$ is any smoothing density function that is centered around $x$. This smoothed objective can be interpreted as the NLL objective under a smoothed data distribution (by applying the smoothing function on top of the data). We discuss this in more details in the supplement (Section A). In practice, we find that both Uniform and Gaussian distributions with mean $x$ effectively prevent the bins from collapsing, and use $\zeta(\cdot | x) = \text{Unif}[x - \frac{\lambda}{2}, x + \frac{\lambda}{2})$ or $\zeta(\cdot | x) = \mathcal{N}(x, \lambda^2)$ in all experiments, truncating on the boundaries of the support of $x \in [0, 1)$. By optimizing $\hat{\mathcal{L}}_s$

with uniform target smoothing, the density in blue in Figure 3 converges to a close approximation of the target density.

The smoothed objective might seem intractable with an integral in the inner summation. Fortunately, the form of the conditional $\log p_\theta(\tilde{x} | x_i^{<t})$ with ADACAT's simple density function allows us to evaluate the integral analytically as long as the smoothing density has an analytic cumulative density function (CDF). If $F(\cdot)$ is the CDF of $\zeta$, then the integral can be analytically computed as:

$$\int_x \zeta(x) \log f_{w,h,k}(x) dx$$
$$= \sum_{j=1}^{k} \left[ (F(c_j + p_j) - F(c_j))(\log h_j - \log w_j) \right] \quad (8)$$

where we recall that $c$ is the prefix sum of $w$ as defined previously. Only the bins that intersect with the support of the smoothing density function contribute to this objective.

### 3.1 RELATIONSHIP WITH DENOISING SCORE MATCHING

Energy based models and denoising autoencoders trained by denoising score matching (DSM, Vincent [2011]) minimize a reconstruction objective:

$$\mathcal{L}_{\text{DSM}}(x) = \mathbb{E}_{\zeta(\tilde{x} | x)} \|x - \hat{x}_\theta(\tilde{x})\|_2^2$$

Assuming the observation model $p_\theta(\cdot | \tilde{x}) = \mathcal{N}(\hat{x}_\theta(\tilde{x}), I)$ is a standard Gaussian,

$$\mathcal{L}_{\text{DSM}}(x) = -\mathbb{E}_{\zeta(\tilde{x} | x)} \left[ \log p_\theta(x | \tilde{x}) \right]$$

resembling (6). However, our target smoothed loss is designed to regularize the output conditional distribution, so perturbations are employed on the output space, not the input space, and our generative model is conditioned on clean, unperturbed observations. Recent works introduce multi-scale perturbations [Song and Ermon, 2019], and denoising diffusion probabilistic models reweight a related variational bound for this class of models to improve sample quality [Ho et al., 2020].

### 3.2 RELATIONSHIP WITH DEQUANTIZATION

Other continuous density estimators like normalizing flows trained on discrete data suffer from degenerate solutions if trained naïvely via maximum likelihood (Ho et al. [2019], Sec. 3.1). Flows suffer from a different failure case than non-smoothed ADACAT. Continuous estimators, *e.g.,* a mixture of Dirac $\delta$ functions, can arbitrarily increase density on discrete training data as the empirical distribution is supported on a set with measure zero. Dequantization avoids the problem by adding continuous noise to observed discrete samples [Theis et al., 2015, Hoogeboom et al., 2021]. As an

example, the following dequantized objective gives a lower bound of the log-likelihood of discrete data sample $x$:

$$\log p_\theta^{\text{DQ}}(x) = \mathbb{E}_{\zeta(\tilde{x}|x)} \log p_\theta(\tilde{x}) \qquad (9)$$

$$= \int_x^{x+\lambda} \zeta(\tilde{x}|x) \log p_\theta(\tilde{x}) d\tilde{x}$$

$$\leq \log \int_x^{x+\lambda} \zeta(\tilde{x}|x) p_\theta(\tilde{x}) d\tilde{x} = \log P_\theta(x),$$

where $\lambda$ is chosen such that $[x, x + \lambda)$ with different discrete sample $x$ do not overlap with each other. While (9) closely resembles (7), it differs subtly in that dequantization perturbs all dimensions of the data $x$, not just the 1-D target, and that the integral is done via a stochastic sample from $\zeta$ rather than analytically. We observe bin collapse even on continuous data like the mixtures in Figures 2, 3, and find that single-sample estimates of the expectation do not prevent collapse. These findings suggest that analytic target smoothing helps with the discontinuity in the model conditional rather than a property of the data.

## 4 EVALUATION

In experiments, we evaluate the performance of autoregressive density estimators with adaptive categorical conditional distributions for several data modalities. We evaluate on standard benchmarks for real-world tabular data (Section 4.1), image generation (4.2), speech synthesis (4.3) and offline reinforcement learning (4.4). ADACAT outperforms uniform discretization strategies in all settings, and is competitive with hand-engineered conditional distributions. Beyond density estimation, our results suggest that ADACAT can improve downstream task performance, including speech quality and control.

### 4.1 TABULAR DATA MODELING

We compare the performance of autoregressive models with ADACAT and uniform parameterizations on real-world tabular density estimation benchmarks, the UCI datasets of Dua and Graff [2017]. The state-of-the-art performances on these benchmarks are also included for reference. We use a 4-layer feed-forward network to predict the ADACAT parameters for each dimension of the data (*e.g.*, we use 6 MLPs for the POWER dataset since it has 6-dimensional data). Each network has 500 hidden units for all datasets except for GAS, where we use 1000 hidden units.

We also use a Fourier encoding of the input inspired by Tancik et al. [2020], Kingma et al. [2021] to allow a shallow model to capture high-frequency variations in the input. Specifically, we augment each input element $x^t$ with $b$ pairs of additional features: $\{\sin(2^j x^t), \cos(2^j x^t)\}_{j=0}^{b-1}$. We choose the feature count $b = 32$ for GAS and POWER, $b = 8$ for MINIBOONE, and $b = 4$ for HEPMASS.

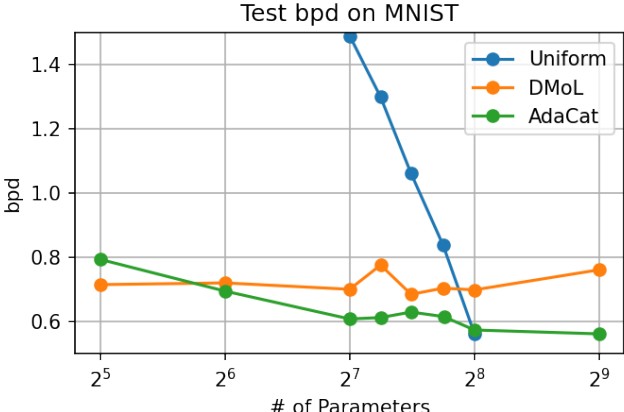

Figure 4: Test bits per dimension (bpd) on MNIST image generation task with different output parameter count. The parameter count is in log scale. The MNIST data is discrete with $2^8 = 256$ possible values for the intensity at each pixel.

The uniform baseline uses the same architecture except the widths of the bins are forced to be uniform. We search over the number of bins in $\{100, 200, 300, 500, 1000\}$ for both the uniform baseline and ADACAT and select the best to report in the table. All models are trained for 400 epochs using Adam [Kingma and Ba, 2014] with a learning rate of 0.0003 and the learning rate halves every 100 epochs. We use truncated Gaussian target smoothing for ADACAT with $\lambda = 0.00001$ for POWER, and $\lambda = 0.0001$ for all other datasets. See more details in the supplement (Section B).

Table 1 reports results. Overall, ADACAT consistently outperforms the uniform baseline across all datasets, reducing the NLL by 1.9, 4.8, 3.1 and 4.0 nats on POWER, GAS, HEPMASS and MINIBOONE, respectively. Our approach also obtains competitive performance with the state-of-the-art normalizing flow models on GAS.

### 4.2 IMAGE DENSITY ESTIMATION

Table 2 compares the performance of ADACAT against existing parameterizations on the grayscale MNIST [LeCun and Cortes, 2010] image generation task in terms of negative log-likelihood measured in bits/dimension. The autoregressive architecture we use for this task is a GPT-like Transformer decoder Vaswani et al. [2017] with 4 layers and 4 heads.[2] Since the image data is discrete, instead of dequantizing and smoothing the target, we directly minimize the cross entropy loss in the original discrete space. We compute the log probability of the $i^{\text{th}}$ discrete target by analytically com-

---

[2]We use the implementation, training pipeline, and the default training hyperparameters from `github.com/karpathy/minGPT`, and treat an image as a token sequence with a vocabulary size of 256. We also use a smaller batch size of 20. See more details in the supplement (Section C.2).

| Method | POWER (m=6) | GAS (m=8) | HEPMASS (m=21) | MINIBOONE (m=43) |
|---|---|---|---|---|
| MADE [Germain et al., 2015] | 3.08 | −3.56 | 20.98 | 15.59 |
| MAF [Papamakarios et al., 2017] | −0.24 | −10.08 | 17.70 | 11.75 |
| NAF-DDSF [Huang et al., 2018] | −0.62 | −11.96 | 15.09 | 8.86 |
| TAN [Oliva et al., 2018] | −0.48 | −11.19 | 15.12 | 11.01 |
| FFJORD [Grathwohl et al., 2019] | −0.46 | −8.59 | 14.92 | 10.43 |
| Block NAF [De Cao et al., 2020] | −0.61 | −12.06 | 14.71 | 8.95 |
| DDEs [Bigdeli et al., 2020] | −0.97 | −9.73 | 11.3 | 6.94 |
| nMDMA [Gilboa et al., 2021] | −1.78 | −8.43 | 18.0 | 18.6 |
| Uniform Discretization | 1.34 | −6.29 | 21.37 | 16.93 |
| ADACAT | −0.56 | −11.27 | 18.17 | 14.14 |

Table 1: **(Tabular Data)** Test negative log-likelihood for density estimation on UCI datasets [Dua and Graff, 2017]. We followed the same data pre-processing pipeline as in Papamakarios et al. [2017]. ADACAT achieves competitive performance on GAS and consistently outperforms the uniform baseline.

| Parameters | Uniform | Adaptive Quantile | DMoL | AdaCat |
|---|---|---|---|---|
| 512 | N/A | × | 0.761 | **0.561** |
| 256 | **0.561** | × | 0.698 | 0.573 |
| 216 | 0.838 | × | 0.704 | **0.615** |
| 180 | 1.061 | × | 0.684 | **0.629** |
| 152 | 1.299 | × | 0.776 | **0.612** |
| 128 | 1.490 | × | 0.700 | **0.608** |
| 64 | 2.453 | × | 0.720 | **0.695** |
| 32 | 3.392 | 1.276 | **0.715** | 0.793 |
| Best | **0.561** | 1.276 | 0.715 | **0.561** |

Table 2: **(Image Generation)** Test negative log-likelihood in bits per dimension (bpd) on MNIST image generation task with different output parameter count. ADACAToutperforms other baselines on most parameter counts. The adaptive quantile baseline diverges with a parameter count higher than 32, indicated by ×.

puting the total probability mass assigned to $\left[\frac{i}{256}, \frac{i+1}{256}\right]$ in our continuous distribution, *i.e.* density integrated over the interval. This corresponds to mapping our continuous distribution from $[0, 1]$ to the 256 discrete values uniformly such that the $i^{\text{th}}$ discrete value is mapped from $\left[\frac{i}{256}, \frac{i+1}{256}\right]$.

The results are grouped according to the number of parameters used to express the intensity distribution of each pixel, allowing us to examine the parameter efficiency of each approach. For the uniform baseline, we evenly divide the $[0, 1]$ intensity interval into $k$ bins and use $k$ parameters to model the probability assigned to each bin (with unnormalized log probability). DMoL uses $3k$ parameters for a $k$-component mixture model (*e.g.*, a 256 parameter count budget for DMoL corresponds to a 86-component mixture model). ADACAT uses $2k$ parameters for a $k$-component mixture model, using $k$ bins of variable size. We examine parameter counts ranging from 32 to 512.

Overall, ADACAT has better performance on most param-

eter counts. It only underperforms the uniform discretization at 256 parameters. We note that the MNIST dataset is discrete with 256 classes, which means that the uniform discretization has the optimal bin division. Therefore, we do not expect ADACAT to be able to outperform uniform because ADACAT has effectively half of the bins available. DMoL achieves the best performance when there are 32 parameters, but scales poorly with the number of components and underperforms ADACAT for most settings with more than 32 parameters.

We also experiment with an adaptive quantile baseline where we keep the probability mass assigned to each bin to be the same rather than the width. However, we found that the adaptive quantile baseline is very unstable to train. We only report its result on a parameter count of 32 because the model at a higher parameter count often diverges early in training which results in inconsistent performances across runs. The adaptive quantile baseline outperforms uniform with 32 parameters, yet is still much less expressive than ADACAT.

### 4.3 AUDIO DENSITY ESTIMATION AND VOCODING

Neural vocoders synthesize human-like speech expressed as a waveform, conditioned on phonetic or frequency spectrum based features. Speech waveforms are long sequences as audio must be sampled at a high rate for fidelity, typically 16-24 kHz. Thus, efficient generative models are essential for practical applications, and the output layer of the network can be a significant fraction of the compute. WaveNet [Oord et al., 2016] is a popular autoregressive vocoder that synthesizes waveforms conditioned on a Mel-spectrogram representation of the amplitude of audio frequencies across multiple bands. Despite the conditioning information, vocoding is challenging as WaveNet needs to reconstruct the phase of the audio frequencies. This is done by estimating the distribution of audio in a dataset via maximum likelihood.

| Conditional dist. | Transform | Parameters | NLL (raw) ↓ | NLL ($\mu$-law 256) ↓ | MCD ↓ |
|---|---|---|---|---|---|
| Gaussian | – | 2 | $-8.39 \pm 0.11$ | $-4.78 \pm 0.08$ | $3.08 \pm 0.02$ |
| Uniform Categorical | – | 30 | $-3.78 \pm 0.01$ | $0.63 \pm 0.01$ | – |
| Uniform Categorical | $\mu$-law | 30 | $-7.71 \pm 0.07$ | $-3.32 \pm 0.03$ | $17.00 \pm 0.49$ |
| DMoL, 10 components | – | 30 | $-8.45 \pm 0.11$ | $-4.84 \pm 0.09$ | $3.00 \pm 0.01$ |
| Adaptive Cat. (ADACAT) | – | 30 | $-8.30 \pm 0.16$ | $-3.88 \pm 0.10$ | $4.87 \pm 0.04$ |
| Uniform Categorical | – | 256 | $-6.28 \pm 0.04$ | $-0.85 \pm 0.04$ | – |
| Uniform Categorical | $\mu$-law | 256 | $-8.76 \pm 0.10$ | $-4.38 \pm 0.07$ | $3.25 \pm 0.01$ |
| DMoL, 85 components | – | $255^{\dagger}$ | $-8.46 \pm 0.11$ | $-4.85 \pm 0.09$ | $3.01 \pm 0.01$ |
| Adaptive Cat. (ADACAT) | – | 256 | $-8.37 \pm 0.10$ | $-3.99 \pm 0.07$ | $3.02 \pm 0.03$ |
| Uniform Categorical | – | 512 | $-6.92 \pm 0.05$ | $-1.44 \pm 0.05$ | – |
| Uniform Categorical | $\mu$-law | 512 | $-8.12 \pm 0.10$ | $-3.73 \pm 0.07$ | $1.99 \pm 0.01$ |
| DMoL, 171 components | – | $513^{\dagger}$ | $-8.46 \pm 0.11$ | $-4.85 \pm 0.09$ | $3.07 \pm 0.01$ |
| Adaptive Cat. (ADACAT) | – | 512 | $-8.33 \pm 0.11$ | $-3.94 \pm 0.08$ | $2.29 \pm 0.02$ |

Table 3: (**Audio vocoding**) Continuous negative log-likelihood (NLL, in bits/dim) and waveform vocoding MCD error for WaveNet with different parameterizations of conditional distributions on the LJSpeech dataset. $^{\dagger}$Discretized Mixture of Logistics (DMoL) requires 3 parameters per mixture component (weight, mean and log scale), so output parameters are approximately matched to baselines.

We train WaveNet with the standard dilated CNN architecture on the open LJSpeech dataset [Ito and Johnson, 2017] of wav files using an open-source implementation. A ground-truth Mel-spectrogram is extracted and used for conditioning WaveNet. For baselines, we use different parameterizations of the conditional distribution: a uniformly discretized categorical, a categorical discretized by a hand-engineered $\mu$-law strategy that sets bin widths logarithmically with intensity, and a discretized mixture of logistics. All models use 24 convolutional layers and are optimized for 500k iterations with Adam. The learning rate is initially 0.001 and is decayed by half every 200k iterations, with batch size 8. An exponential moving average of model parameters is used for testing. Audio is sampled at 22,050 Hz and windowed into blocks of 1024 samples.

We evaluate the continuous negative log-likelihood (NLL) of test waveforms. NLL is measured in bits/dimension with the waveform scaled to a $[-1, 1]$ amplitude. We also measure the NLL of $\mu$-law transformed data with $\mu = 256$, which is also scaled to $[-1, 1]$.

Following Chen et al. [2021], the Mel Cepstral Distance (MCD) objectively quantifies the perceptual similarity of our synthesized audio and reference audio based on an aligned mean squared error metric [Kubichek, 1993]. Samples from the uniformly discretized model led to numerical instabilities in the open-source library used to compute the MCD metric and have clear auditory artifacts, so the MCD metric is omitted.

Table 3 shows results grouped by the number of parameters output by WaveNet for each conditional. Note that ADA-CAT has half the bins of categorical baselines at the same parameter count due to using two parameter vectors, $w$ and

$h$. Across all settings, ADACAT achieves significantly better negative log-likelihood than uniform discretization: 4.52 bpd lower NLL with 30 output parameters, 2.09 bpd with 256 parameters, and 1.41 bpd with 512 parameters.

ADACAT is also competitive with hand-engineered quantization in the $\mu$-law intensity space, despite not having prior knowledge about humans' logarithmic perception of sound intensity, and without DMoL's instabilities. Still, well-tuned $\mu$-law and DMoL strategies perform well, so the main advantage of ADACAT in the audio domain is capturing most of their performance without human-provided inductive bias. ADACAT and the $\mu$-law transform are complementary, and could be used in concert by learning an adaptive discretization of the heuristically transformed interval.

## 4.4 MODEL-BASED OFFLINE REINFORCEMENT LEARNING

ADACAT's parameterization can also be adopted in the dynamics model of a model-based planner for reinforcement learning tasks. We tested our parameterization with Trajectory Transformer [Janner et al., 2021], a recent work in model-based offline RL that uses a Transformer-based architecture to learn the dynamics model of an environment from a dataset offline, and then uses the model to plan online to produce actions for RL agents. The original Trajectory Transformer architecture discretizes each dimension of states and actions into tokens and uses a one-hot embedding to encode them, similar to how a language model handles vocabulary. Since our discretization is done adaptively with context dependency, continuous inputs are more informative.

| Dataset | Uniform | Quantile | AdaCat |
|---|---|---|---|
| HalfCheetah-Medium | 44.0 ±0.31 | 46.9 ±0.4 | **47.8** ±**0.22** |
| Hopper-Medium | 67.4 ±2.9 | 61.1 ±3.6 | **69.2** ±**4.5** |
| Walker2d-Medium | **81.3** ±**2.1** | 79.0 ±2.8 | 79.3 ±0.8 |

Table 4: **(Offline reinforcement learning)** Normalized scores on three D4RL locomotion (v2) tasks [Fu et al., 2020] using Trajectory Transformer [Janner et al., 2021] with three different discretization methods. ADACAT parameterization performs on par with or better than the uniform and quantile methods used in the original paper. Both mean and standard error over 15 random seeds (5 independently trained Transformers and 3 trajectories per Transformer) are reported, following the protocol in the original paper.

Thus, we minimally modify the architecture by replacing the one-hot embedding layer with a linear layer that takes in a scalar input and outputs its embedding. This modification arguably loses some capacity since it has many fewer parameters than the original architecture. Yet, as we show in our experiments, the gain from the flexibility of our parameterization outweighs the potential capacity reduction. We also reduce the number of bins by a factor of 2 to match the parameter size of the output layer since ADACAT requires $2\times$ more parameters than uniform and non-adaptive quantile-based discretization. We use uniform smoothing with a smoothing coefficient of $\lambda = 0.001$ for the target smoothing objective. We also keep the planning hyperparameters the same as the original work for a fair comparison (except for one hyperparameter on action sampling). See more details on other minor differences between our training and planning procedure compared to the original training and planning procedure in the supplement (Section C.3).

Table 4 shows the performance of the RL agent on three D4RL datasets [Fu et al., 2020] under ADACAT's parameterization and two discrete parameterizations used by Janner et al. [2021]. ADACAT performs better than or on par with the uniform and quantile parameterizations used in the original paper, improving return by 2.9% and 5.2% on average, respectively. This demonstrates its effectiveness in accurately modeling continuous data, and the downstream benefits of more expressive discretization.

## 5   RELATED WORK

**Adaptive Discretization**   The idea of adaptive discretization has found tremendous applications in different fields such as reinforcement learning [Chow and Tsitsiklis, 1991], finite element analysis [Liszka and Orkisz, 1980] and computer graphics [Jevans and Wyvill, 1988]. We bring this powerful idea into density modeling by introducing ADACAT. Unlike most existing works on discretization that rely on heuristics and prior knowledge of the data domain [Tang

and Agrawal, 2020, Ghasemipour et al., 2021], ADACAT can be jointly optimized with the rest of the network parameters and learns to adaptively discretize. Bhat et al. [2021] (AdaBins) is closest to our work. Though the idea is similar, AdaBins is different from ADACAT in several important aspects. AdaBins parameterizes the bin widths directly while ADACAT parameterizes the unnormalized log bin widths. AdaBins uses additional regularization loss that encourages the bin centers to be close to the data while ADACAT does not need any additional regularization. AdaBins is primarily used in depth prediction in vision, whereas our work focuses on generative modeling across multiple domains.

**Efficient softmax**   Language models often have vocabulary sizes of 10k-100k tokens. Computing the softmax normalizer for such a large vocabulary can be expensive, motivating more efficient softmax variants, surveyed by Ruder [2016]. Hierarchical softmax [Morin and Bengio, 2005] is one variant that groups tokens in a tree structure, and does not rely on data being ordinal. However, unlike ADACAT, hierarchical softmax groups a fixed, discrete vocabulary, rather than supporting continuous intervals.

**Image density estimation**   Until recently [Kingma et al., 2021], autoregressive models were the-state-of-the-art on image density estimation benchmarks, and are still widely used. Order agnostic models Uria et al. [2014] improve the flexibility of autoregressive models in downstream tasks like inpainting and outpainting [Jain et al., 2020], but do not change the form of the conditional distribution.

**Audio synthesis**   Likelihood-based models are popular in text-to-speech. Efficiency is a key concern, motivating parameter-efficient conditional distributions. Paine et al. [2016] cache intermediate WaveNet activations for improved speed, and Oord et al. [2018] distill WaveNet into a parallel flow. Tacotron [Wang et al., 2017] performed end-to-end speech synthesis from text with a WaveNet vocoder, so our work can be applied to text-to-speech (TTS) systems. Other approaches include Flow-based models, diffusion [Kong et al., 2021, Chen et al., 2021] and GANs [Kumar et al., 2019].

## 6   LIMITATIONS

One fundamental limitation of ADACAT is that it can never model more modes than the number of bins. One possible solution is to add flow steps [Dinh et al., 2016, Rezende and Mohamed, 2015] to transform the space and run the autoregressive model with ADACAT in the transformed space. It is possible that using ADACAT in a transformed or latent space can lead to a more parameter-efficient way of modeling continuous distributions.

Another limitation of ADACAT is that the parameterization

is discontinuous, with more discontinuities as the number of bins increases. We largely resolved the issue using target smoothing. It is possible to use non-uniform density within each bin like spline parameterizations [Durkan et al., 2019], but our preliminary experiments suggested that these are challenging to optimize in practice.

# 7 CONCLUSION

Likelihood-based autoregressive generative models can estimate complex distributions of high-dimensional real-world data, but often struggle with efficiency and can be difficult to expressively parameterize on continuous data. In this paper, we presented ADACAT, a flexible, efficient, multimodal parameterization of 1-D conditionals with applications in autoregressive models. We demonstrated the effectiveness of ADACAT on a diverse set of tasks: image generation, tabular density estimation, audio vocoding and model-based planning. ADACAT improves density estimation and downstream task performance over uniform discretization, and is competitive with other hand-engineered discretizers.

**Acknowledgements**

We would like to thank Michael Janner, Xinyang Geng and Hao Liu for helpful discussions at the early stage of this work. We also thank Sergey Levine for insightful feedback on the offline RL experiments. We thank Katie Kang, Dibya Ghosh, and other members of the RAIL lab for feedback on paper drafts. Qiyang Li is supported by Berkeley Fellowship.

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
