# OpenReview forum: "AdaCat: Adaptive Categorical Discretization for Autoregressive Models"
_auai.org/UAI/2022/Conference — UAI 2022 Poster_

### Official Review · Reviewer_3H8k · 2022-04-11

**Q2(1) Originality/Novelty:** 3
**Q2(2) Significance/Impact:** 4
**Q2(3) Correctness/Technical Quality:** 3
**Q2(6) Clarity Of Writing:** 4
**Q6 Overall Score:** 8
**Q8 Confidence In Your Score:** 3

**Q1 Summary And Contributions:**

The paper considers the problem of quantization of continuous distribution. This is an important problem in ML since many models (e.g., deep generative models) require a dequantization of a continuous distribution for discrete random variables. Instead, the authors propose a new class of distributions, Adaptive Categorical Discretization, to deal with the problem. The paper is very interesting!


**Q2 Assessment Of The Paper:**

More detailed information regarding each of these aspects is given below:

**Q2(4) Quality Of Experiments (Optional):**

3: Good: The experimental evaluation is adequate, and the results convincingly support the main claims.

**Q2(5) Reproducibility:**

3: Good: Key resources (e.g., proofs, code, data) are available and key details (e.g., proofs, experimental setup) are sufficiently well-described for competent researchers to confidently reproduce the main results.

**Q3 Main Strengths:**

+ The considered problem is very important for the ML community and vital in deep generative modeling.
+ The proposed approach is neat and simple but in the best possible sense of this word! It is one of these ideas that you think that how could we live without it.
+ The experiments are very convincing.

**Q4 Main Weakness:**

I do not see any main weaknesses. For some minor points please see Q5.

**Q5 Detailed Comments To The Authors:**

- It would be interesting to take a look at more dequantization techniques:
Hoogeboom, E., Cohen, T., & Tomczak, J. M. (2020, November). Learning Discrete Distributions by Dequantization. In Third Symposium on Advances in Approximate Bayesian Inference. [https://openreview.net/forum?id=a0EpGhKt_R]
- Figure 4 (caption): parmaeter -> parameter

**Q7 Justification For Your Score:**

In my opinion, the paper is extremely well written and the idea is very neat! The experimental part is convincing.

**Q9 Complying With Reviewing Instructions:**

1: Yes.

---

### Official Review · Reviewer_ScDq · 2022-04-12

**Q2(1) Originality/Novelty:** 2
**Q2(2) Significance/Impact:** 3
**Q2(3) Correctness/Technical Quality:** 3
**Q2(6) Clarity Of Writing:** 3
**Q6 Overall Score:** 5
**Q8 Confidence In Your Score:** 3

**Q1 Summary And Contributions:**

The paper proposes discretizing each dimension of an autoregressive model adaptively, which enables the model to focus on intervals of interest. The experiments show that the proposed method improves density estimation on various applications including tabular data, images and audio.

**Q2 Assessment Of The Paper:**

More detailed information regarding each of these aspects is given below:

**Q2(4) Quality Of Experiments (Optional):**

3: Good: The experimental evaluation is adequate, and the results convincingly support the main claims.

**Q2(5) Reproducibility:**

2: Fair: Key resources (e.g., proofs, code, data) are unavailable but key details (e.g., proof sketches, experimental setup) are sufficiently well-described for an expert to confidently reproduce the main results.

**Q3 Main Strengths:**

1) The papers proposes a simple and intuitive approach to improve the density estimation.
2) The experiments are thorough and investigate various applications.
3) The approach is shown to be effective empirically.


**Q4 Main Weakness:**

1) As adaptive discretization is not novel, I still think the paper is incremental, even if the application is novel.
2) The writing can be improved to be clearer.
3) The authors provided one possible explanation for mode collapse, but did not provide a conclusive argument.
4) The smoothed objective solution appears to be ad-hoc.


**Q5 Detailed Comments To The Authors:**

The paper can be improved by providing a more rigorous theoretical analysis, as described above.

**Q7 Justification For Your Score:**

The paper is well-written, and while incremental, has provided thorough empirical results showing the effectiveness of the proposed approach. There is room for improvement in terms of more rigorous theoretical results and explanations, regarding the local optima.

**Q9 Complying With Reviewing Instructions:**

1: Yes.

---

### Official Review · Reviewer_ztra · 2022-04-14

**Q2(1) Originality/Novelty:** 3
**Q2(2) Significance/Impact:** 3
**Q2(3) Correctness/Technical Quality:** 3
**Q2(6) Clarity Of Writing:** 4
**Q6 Overall Score:** 7
**Q8 Confidence In Your Score:** 4

**Q1 Summary And Contributions:**

This paper proposes Adaptive Categorical Discretization (ADACAT) to tackle the problem of using categorical distributions over the bins to approximate the continuous data distribution. ADACAT discretizes each dimension of an autoregressive model adaptively, which allows the model to allocate density to fine intervals of interest, improving parameter efficiency.

**Q2 Assessment Of The Paper:**

More detailed information regarding each of these aspects is given below:

**Q2(4) Quality Of Experiments (Optional):**

3: Good: The experimental evaluation is adequate, and the results convincingly support the main claims.

**Q2(5) Reproducibility:**

3: Good: Key resources (e.g., proofs, code, data) are available and key details (e.g., proofs, experimental setup) are sufficiently well-described for competent researchers to confidently reproduce the main results.

**Q3 Main Strengths:**

ADACAT is based on the observation that high precision is often only required in a small subset of a distribution’s support. Specifically, The ADACAT distribution is a particular subfamily of mixtures of uniform distributions where each mixture component has non-overlapping support. Second, the author proposed a smoothed objective to alleviate the problem of poor local optima by maximizing the empirical log likelihood. Third, the authors discuss the relations of ADACAT and denoising score matching, ADACAT and dequantization. The authors evaluate this method on tabular data, image data, audio data and model based off-line reinforcement learning. Lastly, this paper is well and clearly written.

**Q4 Main Weakness:**

The author evaluates the ADACAT on MNIST dataset which are relatively small images. It would be interesting to see how ADACAT would work on large images, given most generative models (except GANs) cannot handle large images.

**Q5 Detailed Comments To The Authors:**

See above.

**Q7 Justification For Your Score:**

The problem they try to address is interesting but it has limited applications given the other methods that exist.

**Q9 Complying With Reviewing Instructions:**

1: Yes.

---

### Decision · Program_Chairs · 2022-05-15

**Decision:**

Accept (Poster)

**Comment:**

Meta Review: Reviewers mostly found the method to be novel, simple, and useful. It addresses an important problem in the realm of density estimation, and shows convincing experimental results on a very diverse set of problems. In addition, the writing and presentation are clear and compelling. During the discussion the authors derived some further results which I trust will be incorporated into the final accepted paper (possibly in the supplemental).